# Effects of Combining Different Collaborative Learning Strategies with Problem-Based Learning in a Flipped Classroom on Program Language Learning

**Yi-Hsing Chang ***, **Yin-Chen Yan and You-Te Lu**

Department of Information Management, Southern Taiwan University of Science and Technology, Tainan 71005, Taiwan; yinzoe@stust.edu.tw (Y.-C.Y.); yowder@stust.edu.tw (Y.-T.L.)
* Correspondence: yhchang@stust.edu.tw

**Abstract:** This study proposed combining problem-based learning (PBL) with different collaboration learning strategies in flipped classrooms to improve learners' learning motivation and learning outcomes. The main idea was to design a teaching process based on the concept of flipped classrooms. The proposed method was adopted to design learning objectives, learning content, and group activities, thereby forming new teaching strategies for developing students' independent learning, logical thinking, problem-solving skills, learning outcomes, and learning motivation. We used the C# programming language as the learning target and recruited 96 students from a first-year class of the Department of Information Management of the Southern Taiwan university of Science and Technology as participants. During the experiment, the participants were divided into an experimental group with heterogeneous subgroups and a control group with random subgroups, and pretest and posttest data obtained during the programming course were used to evaluate the learning effects of the proposed teaching strategies. In addition, a questionnaire was used to explore learning motivation through three aspects: flipped classroom, PBL, and collaborative learning. The results of this study indicated that the proposed teaching strategies improved the participants' learning outcomes. The experimental group exhibited higher improvements in learning outcomes than did the control group. Significant results were obtained for all the items of the adopted questionnaire. Thus, the participants provided positive ratings to the flipped classroom model designed in this study.

**Keywords:** flipped classroom; problem-based learning; collaboration learning; programming language



## 1. Introduction

Taiwan implemented a new set of curriculum guidelines (i.e., Curriculum Guidelines of 12-year Basic Education) starting in the 2019–2020 academic year. According to the 11th item of the curriculum guidelines, teaching in the field of science and technology is aimed at cultivating students' higher-order thinking skills, including creative thinking, logical and computational thinking, critical thinking, and problem-solving. To cultivate these skills, computer programming skills have become essential information skills for middle school, high school, and university students. The learning of programming can cultivate students' problem-solving and high-level thinking abilities, and programming skills can bring positive benefits to future career planning [1].

The most difficult aspect of programming is that the manner in which humans solve problems is completely different from that in which computers describe them. The syntactic structure of general programming is complex and large, and beginners find it difficult to learn programming only through self-learning [2]. The traditional programming teaching method emphasizes the importance of syntax. Beginners often fall into the trap of using a trial-and-error method when creating a program [3]. This phenomenon often leads to the deepening of students' frustration in learning programming because of the mistakes that they make during the learning process, which in turn reduces their willingness to learn.

Mazur [4] indicated that teachers do not consider individual differences between students and pay excessive attention to the transfer of knowledge. Thus, teachers tend to ignore the processes through which students absorb and internalize knowledge, which makes it difficult to arouse students' learning motivation. Demir [5] investigated the effect of educational programming language integration on academic achievement and programming anxiety level. The research concluded that educational programming languages can be used by integrating both the theory and practice of the course to increase academic success and in-class performance and reduce anxiety about computer programming.

In recent years, considerable research has been conducted on the application of flipped classrooms in programming courses. GökçeAkçayır and MuratAkçayır [6] presented a large-scale systematic review of the literature on the flipped classroom, with the goals of examining its reported advantages and challenges. The findings reveal that the most frequently reported advantage of the flipped classroom is the improvement of student learning performance. Fetaji et al. [7] evaluated the effects of applying a flipped classroom model in computer science courses on students' learning outcomes. They analyzed the influences of the flipped classroom concept through empirical research and found that the application of the flipped classroom model had a positive influence on the students' learning outcomes. The application of the aforementioned model increased opportunities for teacher–student communication, and online teaching materials were used to fill the gaps in teacher–student communication and improve learning outcomes. Elmaleh [8] discussed the effects and student acceptance of the application of a flipped classroom model in programming courses. Their results indicated that compared with traditional classrooms, flipped classrooms resulted in improvements in students' final exam passing rates, learning outcomes, and problem-solving skills. Yan [9] also explored the effects and student acceptance of a flipped classroom model in programming courses. Their results indicated that flipped classrooms can increase students' acceptance of programming. Knutas [10] not only examined the application of flipped classroom model in university programming courses but also established a learning framework by using the curriculum design of flipped classrooms. Knutas also found that flipped classrooms are more effective than traditional learning methods and suggested that teachers incorporate the concept of flipped classrooms into their teaching design. Maher [11] explored the differences in motivation, independent learning, programming attitude, and learning outcomes generated through flipped classrooms and traditional teaching. They found that students provided positive feedback on the implementation of the flipped classroom model. Furthermore, these methods aroused learning motivation and promoted active learning.

According to the results of relevant studies, the flipped classroom model can improve the disadvantages of traditional teaching methods. However, this method also has many disadvantages. For example, the flipped classroom learning design must be revised for truly enhancing learning motivation [12]. Alhazbi [13] explored the applicability of the flipped classroom model in programming courses and found that this model can improve learning attitude and performance in programming courses; however, attention must be paid to how to encourage students to preview class content and prepare for a class in advance. Taşpolat, Özdamli, and Soykan [14] determined the impact of the flipped classroom approach on students' academic achievement and their attitudes toward programming and methodology at the higher education level. The research found that this method also has many disadvantages, such as the need for technological requirements, students not watching videos, poor attendance to the course, and lowered student–teacher interaction, especially outside the classroom. GökçeAkçayır and MuratAkçayır [6] also found a number of challenges, the majority of these related to out-of-class activities, such as much reported inadequate student preparation prior to class. Hendrik and Hamzah [15] presented a systematic literature review of the flipped classroom approach in the programming course. The research found that most in-class activities related to practical activities instead of active learning activities. Since the programming course requires students to have more practice time to master the skill, the authors suggest that the teachers should consider

the activity as involving a more active process. Although flipped classrooms can enhance learners' learning outcomes, teachers are often unable to understand students' internal problems and solve them in a timely manner during the teaching process; thus, they do not receive positive feedback in terms of cooperative learning and programming attitude. Consequently, to overcome the problems associated with flipped classroom learning, it must be supplemented with supporting teaching measures.

Collaborative learning signifies specific group learning in which students work together. In addition to maintaining their individual contributions to their groups, students work together with other group members to achieve a common goal. According to Johnson and Johnson [16], the spirit of collaborative learning lies in the idea that a group's learning is truly successful only when each member achieves common goals. Therefore, true collaborative learning requires mutual support, assistance, sharing, and encouragement among team members. Each member in a group has certain teaching and learning responsibilities, and group members must teach and learn from each other to achieve the group's common goals.

Problem-based learning (PBL) is an effective teaching method that involves teaching students to learn from problems. PBL originated in the field of medical education in the United States for cultivating talents. In PBL, "problems" are used as the materials to motivate learning, and students follow the problem framework to explore and learn in order to construct a knowledge base. Subsequently, students are guided to learn actively through problem solving. In PBL, teachers guide students to form case studies based on real-life problems, discuss these case studies in groups, and propose solutions to the problems. Moreover, appropriate teaching methods are designed according to the course content, and students are guided to apply the course knowledge learned before class so that their learning is not limited to simply reading rigid textbooks but includes active learning and the application of learned knowledge to real-life problems.

Therefore, to enhance learners' participation and interaction in the classroom, we incorporated PBL and different collaborative learning strategies into flipped classrooms to further enhance learners' learning outcomes and motivation.

We investigated the following questions under the implementation of the aforementioned flipped classroom model:

- In a flipped classroom, can the combination of PBL and different collaborative learning strategies enhance learners' learning motivation?
- In a flipped classroom, can the combination of PBL and different collaborative learning strategies enhance learners' learning outcomes?

## 2. Methodology

### 2.1. Research Architecture

The research architecture is as shown in Figure 1, containing the control variable, independent variable, and dependent variable.

The research framework of this study is shown in Figure 1. We investigated whether the application of flipped classrooms (the main learning model) combined with PBL and different collaborative learning strategies could enhance learners' learning outcomes and motivation in a programming course.

(1) Control variable:
- Flipped classroom: An experimental group and a control group were taught using the flipped classroom model.
- Teaching content: The teaching content for the experimental and control groups were the same.
- Problem-based learning (handout): Worksheets were used for learning.
- Collaboration: The experimental and control groups were divided into subgroups in group activities.

- Instructor: The experimental and control groups were taught by the same instructor.

(2) Independent variable:

- Grouping: The control group was randomly divided into subgroups, whereas the experimental group was divided into heterogeneous subgroups according to previous test scores.

(3) Dependent variable:

- This study had three objectives: to examine whether the adopted teaching method improved students' learning outcomes, to compare the difference in learning outcomes between the experimental and control groups, and to investigate the effect of the adopted teaching method on the students' motivation in program language learning.

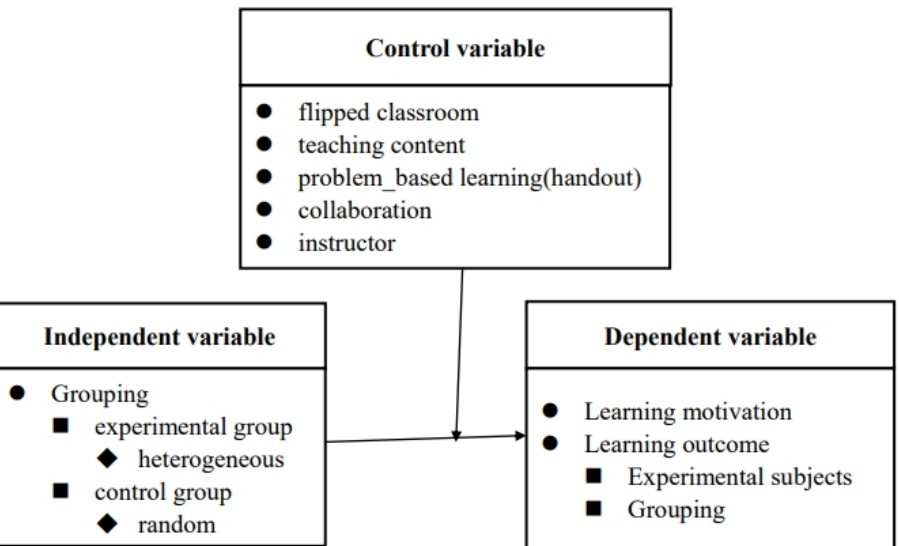

**Figure 1.** Research framework.

*2.2. Experiment Design*

2.2.1. Experiment Duration and Procedure

(1) Experiment: Three 50-min classes were conducted per week (total weekly learning time of 150 min) for 9 weeks in one semester. PBL and collaborative learning strategies were incorporated into these classes. The implementation steps of the 9-week course were as follows:

Step 1. During the course, the instructor explained the learning topic and learning content for 50 min.
Step 2. The students practiced using lesson examples for 40 min.
Step 3. The students were provided a worksheet to solve problems through collaborative learning, and they had to complete and upload their solutions to a learning platform before the end of the course (within 60 min).

(2) Post-experiment questionnaire: After the course, the experimental group filled a post-experiment questionnaire on the learning platform.

2.2.2. Learning Achievement and Questionnaire Evaluation

An independent samples *t*-test was used to assess the learning achievement of learners. In addition, the adopted questionnaire was completed using a 5-point Likert scale with the following scoring options: 5, 4, 3, 2, and 1 stood for "strongly agree", "agree", "neither agree or disagree", "disagree", and "strongly disagree", respectively.

2.2.3. Research Hypotheses

On the basis of the research questions, the following hypotheses were formulated:

**Hypothesis 1 (H1).** *The experimental subjects (control group + experimental group) would exhibit significant differences in learning outcomes after the implementation of the proposed teaching strategy.*

**Hypothesis 2 (H2).** *Significant differences would exist between the learning outcomes of the experimental and control groups after the implementation of the proposed teaching strategy.*

**Hypothesis 3 (H3).** *Significant differences would exist between the learning outcomes of the high-scoring students of the experimental group before and after the implementation of the proposed teaching strategy.*

**Hypothesis 4 (H4).** *Significant differences would exist between the learning outcomes of the medium-scoring students of the experimental group before and after the implementation of the proposed teaching strategy.*

**Hypothesis 5 (H5).** *Significant differences would exist between the learning outcomes of the low-scoring students of the experimental group before and after the implementation of the proposed teaching strategy.*

**Hypothesis 6 (H6).** *The proposed teaching method would improve the learning motivation of the learners.*

2.2.4. Research Instruments

Three research tools were used in this study: a computer, a digital learning platform, and statistical software.

- Computer: The experiment site was a computer classroom. Each student was provided a computer, which they used for online searches, programming practice, and the programming test.
- Digital learning platform: The FLIP Digital Learning Platform of Southern Taiwan University of Science and Technology was used for providing the students learning materials and resources (https://flipclass.stust.edu.tw, accessed on 2 March 2022).
- Statistical software: The collected data were statistically analyzed using IBM SPSS Statistics V25.

## 3. Teaching Design

### 3.1. Teaching Environment

The teaching environment is shown in Figure 2. By using the basic functions provided by the FLIP platform, such as textbooks, group zone, questionnaires, assignments, tests, and discussions, as well as Visual Studio C# on a computer, the learners could conduct independent learning, participate in discussions, work on assignments, submit assignments, take online tests, and fill out questionnaires. The instructor could use the FLIP platform's learning data analysis results to learn about the learners' learning status.

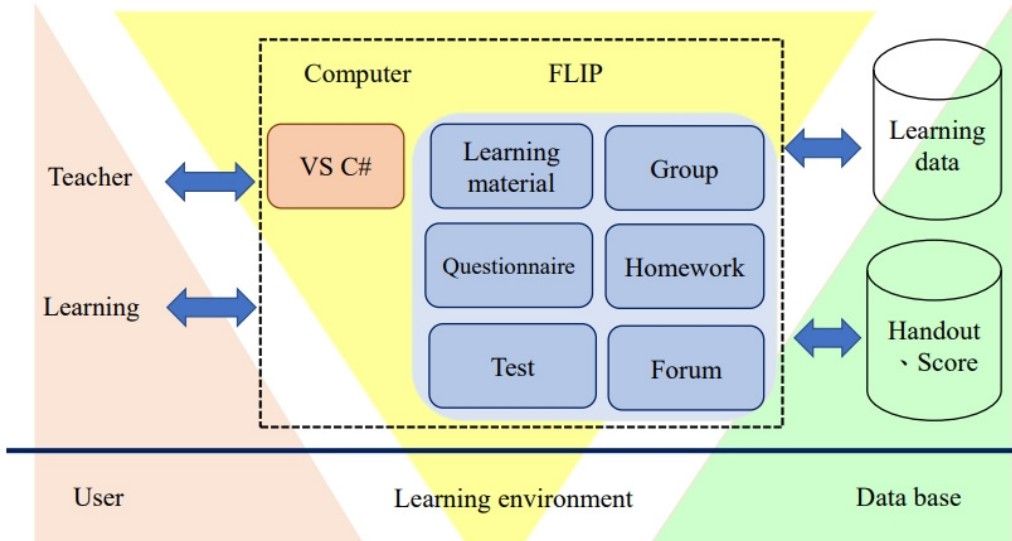

**Figure 2.** Instructional environment.

*3.2. Teaching Flow Design*

The teaching flow design (Figure 3) included a preview before class to understand the content of the unit, in-class explanation of crucial concepts by the instructor, group-based collaborative learning, group activities, and a question and answer session.

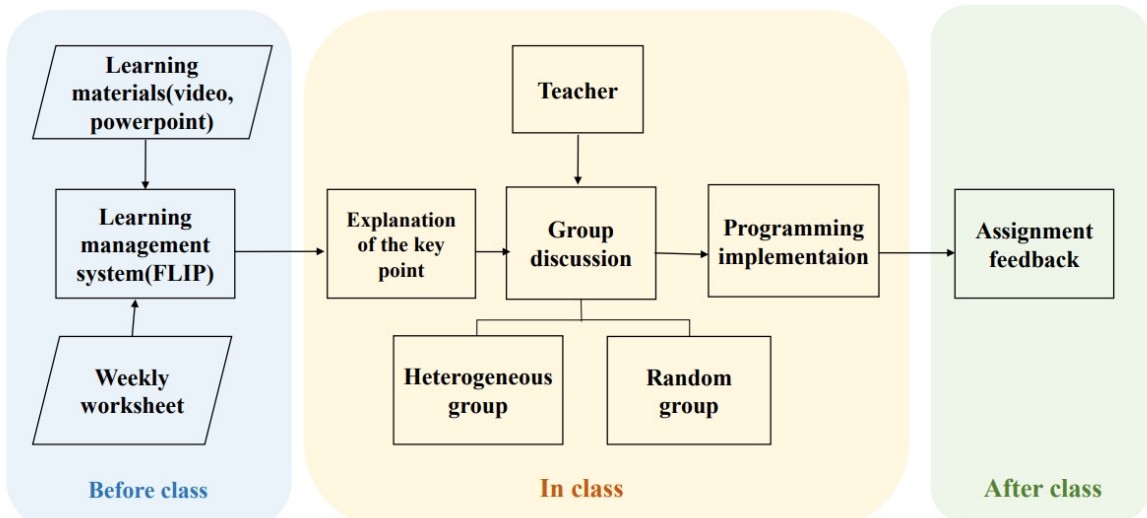

**Figure 3.** Teaching flow design in this study.

(1) Preview before class: The instructor reminded the students to watch teaching videos, slides, and worksheets on the digital learning platform to preview the learning materials for the week.

(2) In-class explanation: First, the instructor explained the learning objectives and key summaries of the week. The instructor used the learning materials to explain the concepts and syntax of the learning content. On the basis of the worksheet, the instructor then allowed the learners to conduct subgroup discussions and self-study and complete the content of the worksheet together. During the subgroup discussion, the learners could ask the instructor questions at any time, and the instructor observed and assisted the students from the sidelines.

(3) After-class activities: The students could visit the digital learning platform to rewatch the assignments completed by their subgroup, review their homework, and engage in discussion and sharing.

*3.3. Design of the Lesson Content and the Handouts*

The learning content was the C# course in the second semester for first-year students of the Department of Information Management of the Southern Taiwan University of Science and Technology. A total of 8 weeks of experimental courses were conducted. The learning topics included arrays and strings, file and folder processing, file reading and writing applications, multimedia files, and databases. The teaching of each topic was completed using audio-visual explanation files and slides (online teaching materials).

This study adopted the concept of PBL, in which questions were used as teaching materials. The learning process was learner-centered, and the worksheets were designed on the basis of concepts such as learning through discussions and group work so that the learners could integrate knowledge when solving questions. Questions were used for guiding the learners' learning and thinking as well as the group discussions and learning. The worksheet mainly contained three parts: teaching objectives, learning content, and learning procedure.

Teaching topics

Year/Month/Day

This section described the topics taught, provided the learners with the basic concepts of the topic, and enabled the learners to understand the learning content quickly.

1. Setting Teaching Goals:
   - The instructor explained to the students that the teaching materials had to be previewed before class to understand the basic concepts.
   - The instructor explained how the course would be conducted and introduced the outline and content of the teaching topics.
   - The instructor explained how the learning activities would be conducted in class and the goals that were expected to be achieved.

2. Learning Content Design

The learning content was explained briefly, themes of the learning questions related to "life practices" were designed, and the directions of the design thinking were presented in a column format to prompt learners to engage in group discussion with a common goal, such that the learning content and teaching topics were synchronized.

   (1) Question
   (2) Thinking direction

3. Learning Procedure

The learning procedure was described in a column format. The learning procedure provided guidance to the learners on how to learn and what to complete.

The following text provides an explanation of the adopted worksheet by using the example of subgroup homework for an array application that involved generating six random integers between 1 and 49.

Array application

2021/09/30

The learners had to understand the basic concepts and array application. An array is a data structure that contains some variables that can be determined by computing indices. The variables contained in an array (also called the elements of an array) are of the same type (also called the element type of the array). An array can be one-dimensional, two-dimensional, three-dimensional, multidimensional, or irregular.

1.  Teaching targets

    (1) Through pre-recorded videos or slides, the students learned what arrays were and the relevant syntax of arrays.
    (2) The learners combined the pre-class preview and in-class learning content to understand the syntax, semantics, and application of arrays.
    (3) The instructor used the worksheet to allow a group to work together so that the students could complete the application program by using arrays.

2.  Learning Content Design

    After a group followed the guidance provided by the instructor to integrate the content of the preview and group discussion, the group used arrays to complete the following application program:

    (1) Question: Picking lottery numbers via a computer (generation of six random integers between 1 and 49), as shown in Figure 4.
    (2) Thinking direction

    - Methods were used on random objects to generate random numbers.
    - The random integer generated by a judgment could not be repeated.
    - The array was completed using the generated random integers.

3.  Procedure

    (1) The students had to understand the basic concepts of arrays through instructional videos.
    (2) The students had to engage in group discussion, flip through textbooks, and search the Internet to obtain the information required for solving a problem.
    (3) The students had to collect data and discuss solutions with others in their group.
    (4) The students had to complete the relevant application on their computer.
    (5) The students had to submit their application online.

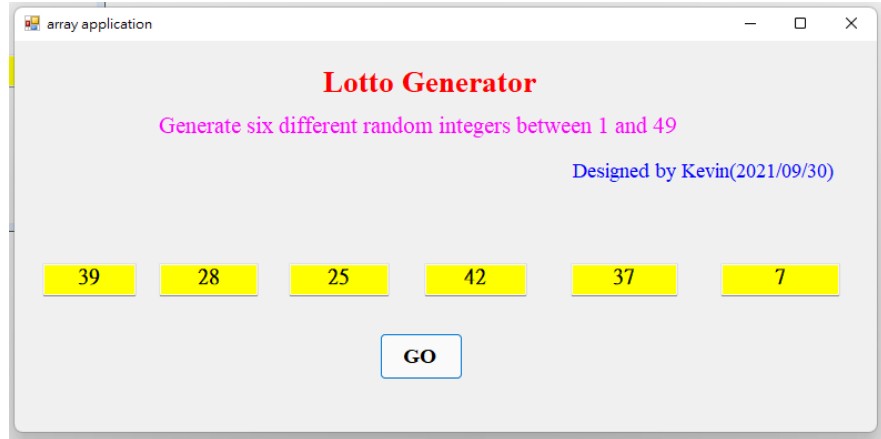

**Figure 4.** Generate six different random integers between 1 and 49.

### 3.4. Grouping Design

The participants of this study were first-year students of the Department of Information Management of our university. The two groups of this study are described as follows:

(1) The experimental group: This group comprised 47 students, who were divided into heterogenous subgroups according to their scores in the midterm programming test. Each subgroup contained one high-scoring student (top 25%), 1–2 medium-scoring students (middle 50%), and one low-scoring student (bottom 25%).

(2) The control group: This group contained 49 members, who arranged themselves into subgroups on their own or were randomly arranged into subgroups by their instructor.

## 4. Experimental Results

An independent samples *t*-test was used to analyze the pretest results of the two groups, before the start of the experiment.

As presented in Table 1, the average pretest scores of the experimental group and the control group were 54.11 and 54.98, respectively. The corresponding *p*-value is 0.791 (>0.05), which does not reach a significant level. Therefore, the groups' basic capability was the same.

**Table 1.** Pretest analysis (independent sample *t*-test analysis).

|  | NO. | Mean | SD | *t* |
|---|---|---|---|---|
| Experimental | 47 | 54.11 | 16.387 | −0.266 |
| Control | 49 | 54.98 | 15.832 | |

*p* > 0.05.

### 4.1. Analysis of Learning Outcomes

4.1.1. Analysis of the Pre- and Post-Test Results of the Experimental Subjects

A paired sample *t*-test is used to evaluate the pre- and post-tests of the experimental subjects. As presented in Table 2, the mean of the pretest and post-test was 54.55 and 57.16, respectively. The corresponding *p* value is 0.001 (<0.01), which does reach a significant level.

**Table 2.** Analysis of the paired samples *t*-test of the experimental subjects.

| Control | NO. | Mean | SD | *t* |
|---|---|---|---|---|
| Pretest | 96 | 54.55 | 16.027 | 1.636 |
| Post-test | 96 | 57.16 | 15.369 | 1.569 |

*p* = 0.001 < 0.01.

We further analyzed the differences in learning outcomes between the experimental group and the control group.

4.1.2. Analysis of the Pre- and Post-Test Results of the Control Group

A paired sample *t*-test was used to evaluate the pre- and post-tests of the control group. As presented in Table 3, the mean of the pretest and post-test were 54.98 and 53.02, respectively. The corresponding *p* value is 0.430 (>0.05), which does not reach a significant level.

**Table 3.** Analysis of the paired samples *t*-test of the control group.

| Control | NO. | Mean | SD |
|---|---|---|---|
| Pretest | 49 | 54.98 | 15.832 |
| Post-test | 49 | 53.02 | 15.250 |

*p* > 0.05.

4.1.3. Analysis of the Pre- and Post-Test Results of the Experimental Group

A paired sample *t*-test was used to evaluate the pre- and post-tests of the experimental group. As presented in Table 4, the mean of the pretest and post-test were 54.11 and 61.47, respectively. The corresponding *p* value is 0.003 (<0.01), which does reach a significant level.

**Table 4.** Analysis of the paired samples *t*-test of the experimental group.

| Experimental | NO. | Mean | SD |
|---|---|---|---|
| Pretest | 47 | 54.11 | 16.387 |
| Post-test | 47 | 61.47 | 14.419 |

*p* < 0.01.

### 4.1.4. Post-Test Analysis

To determine the difference between the experimental and control groups after learning, an independent sample $t$-test was used to evaluate the post-test results of the experimental and control groups. As presented in Table 5, the post-test score of the experimental group was 8.45 points higher than that of the control group. The corresponding $p$ value is 0.006 (<0.01), which does reach a significant level.

**Table 5.** Analysis of the independent samples $t$-tests of the experimental and control groups.

|              | NO. | Mean  | SD     | $t$   |
| ------------ | --- | ----- | ------ | ----- |
| Experimental | 47  | 61.47 | 14.419 | 2.795 |
| Control      | 49  | 53.02 | 15.25  |       |

$p < 0.01$.

### 4.1.5. Analysis of the Pre- and Post-Test Results of Each Group in Experimental Group

To understand the learning outcomes of each category of learners in the experimental group, we conducted a paired sample $t$-test analysis on the pretest and post-test scores for these categories.

As presented in Table 6, the average pretest and post-test scores of the high-scoring students are 69.62 and 67.00, respectively. The corresponding $p$ value is 0.480 (>0.05), which does not reach a significant level.

**Table 6.** Results of the paired sample $t$-test for the pretest and post-test scores of the high-scoring students.

|           | NO. | Mean  | SD     |
| --------- | --- | ----- | ------ |
| Pretest   | 13  | 69.62 | 8.057  |
| Post-test | 13  | 67.00 | 17.445 |

$p > 0.05$.

As presented in Table 7, the average pretest and post-test scores of the medium-scoring students were 56.05 and 60.48, respectively. The relevant $p$ value is 0.112 (>0.05), which does not represent a significant level.

**Table 7.** Results of the paired sample $t$-test for the pretest and post-test scores of the medium-scoring students.

|           | NO. | Mean  | SD     |
| --------- | --- | ----- | ------ |
| Pretest   | 21  | 56.05 | 9.573  |
| Post-test | 21  | 60.48 | 10.529 |

$p > 0.05$.

As presented in Table 8, the average pretest and post-test scores of the low-scoring students were 35.46 and 57.54, respectively. The corresponding $p$ value is 0.000 (<0.001), which indicates statistical significance. Thus, the learning outcomes of the low-scoring students improved significantly after implementing the proposed teaching strategy.

**Table 8.** Results of the paired sample $t$-test for the pretest and post-test scores of the low-scoring students.

|           | NO. | Mean  | SD     |
| --------- | --- | ----- | ------ |
| Pretest   | 13  | 35.46 | 12.959 |
| Post-test | 13  | 57.54 | 16.008 |

$p < 0.001$.

*4.2. Questionnaire Analysis*

A questionnaire was employed for qualitative analysis. A total of 96 copies of the questionnaire were distributed, of which 82 were valid; thus, the valid response rate was 85%. Out of the 82 retrieved questionnaire copies, 42 were retrieved from the experimental group and 40 were retrieved from the control group.

4.2.1. Reliability Analysis

As presented in Table 9, all dimensions received an $\alpha$ value higher than 0.7. The overall scale received an $\alpha$ value of 0.966, implying a certain degree of reliability.

**Table 9.** Reliability analysis of the questionnaire.

| Subscale Name | No. of Items | Cronbach's $\alpha$ |
|---|---|---|
| Flipped classroom | 6 | 0.918 |
| Problem-based learning | 8 | 0.930 |
| Collaboration learning | 7 | 0.944 |
| Total | 21 | 0.966 |

4.2.2. Analysis of Descriptive Statistics

Table 10 shows the results of the questionnaire on the dimension of the flipped classroom. Mean scores on all the items were higher than 3.5, and the grand mean score was 3.815. This result shows that learners achieved a highly satisfactory level for this dimension.

**Table 10.** Flipped classroom.

| NO. | Item | Mean | SD |
|---|---|---|---|
| F1 | The flipped classroom model enhanced my learning motivation. | 3.935 | 0.807 |
| F2 | The flipped classroom model improved my active learning ability. | 4.000 | 0.768 |
| F3 | I previewed the class content before class. | 3.355 | 0.925 |
| F4 | The flipped classroom model improved my programming ability. | 3.935 | 0.866 |
| F5 | Learning through the flipped classroom model allowed me to pass this course. | 3.871 | 0.778 |
| F6 | Overall, I like the flipped classroom model. | 3.790 | 0.890 |
| | Grand mean | 3.815 | 0.839 |

Table 11 shows the results of the questionnaire on the dimension of the problem-based learning. Mean scores on all the items were higher than 3.5, and the grand mean score was 3.948. This result shows that learners achieved a highly satisfactory level for this dimension.

Table 12 shows the results of the questionnaire on the dimension of collaboration. Mean scores on all the items were higher than 3.5, and the grand mean score was 4.053. This result shows that learners achieved a highly satisfactory level for this dimension.

The only difference between the experimental and control groups was the grouping method adopted for them. Therefore, to gain a better understanding of the differences between these groups, we reorganized some of the items from the aforementioned three dimensions into Table 13 to analyze the learning scenarios of the groups. The control group outperformed the experimental group on items P6, C5, and C6, and the experimental group outperformed the control group on items P7, C4, and C7.

**Table 11.** Problem-based learning.

| NO. | Item | Mean | SD |
|---|---|---|---|
| P1 | The worksheets helped me understand my learning goals. | 3.903 | 0.646 |
| P2 | The worksheets helped me understand what I was learning. | 3.871 | 0.665 |
| P3 | Learning by using the worksheets improved my logical thinking skills. | 3.919 | 0.795 |
| P4 | Learning by using the worksheets improved my problem-solving skills. | 3.919 | 0.753 |
| P5 | The guidance provided by the worksheets helped me complete the homework on the worksheet. | 3.903 | 0.694 |
| P6 | Our group could complete the assignments on the worksheets within the stipulated time. | 4.113 | 0.851 |
| P7 | Our group could complete the assignments on the worksheets correctly. | 4.048 | 0.756 |
| P8 | I like learning through worksheets. | 3.903 | 0.783 |
| | Grand mean | 3.948 | 0.743 |

**Table 12.** Collaboration.

| NO. | Item | Mean | SD |
|---|---|---|---|
| C1 | Working in a group helped enhance my learning motivation. | 4.016 | 0.799 |
| C2 | Working in a group helped improve my learning. | 3.984 | 0.799 |
| C3 | Through group work, I could understand the learning content faster. | 3.968 | 0.789 |
| C4 | Our group discussed together to solve the problem. | 3.984 | 0.949 |
| C5 | When we faced a problem, I taught the rest of the group (or a member of the group taught me). | 4.097 | 0.762 |
| C6 | I participated in discussions within the group. | 4.274 | 0.657 |
| C7 | I like to work together as a group to solve problems. | 4.048 | 0.798 |
| | Grand mean | 4.053 | 0.793 |

**Table 13.** Comparison of the results obtained for the two groups.

| NO. | Item | Average | |
|---|---|---|---|
| | | Experimental | Control |
| P6 | Our group could complete the assignments on the worksheets within the stipulated time. | 4.063 | 4.167 |
| P7 | Our group could complete the assignments on the worksheets correctly. | 4.094 | 4.000 |
| C4 | Our group discussed together to solve the problem. | 4.000 | 3.967 |
| C5 | When we faced a problem, I taught the rest of the group (or members of the group taught me). | 4.094 | 4.100 |
| C6 | I participated in discussions within the group. | 4.219 | 4.333 |
| C7 | I like working together as a group to solve problems. | 4.063 | 4.033 |

To examine whether the low-scoring students were less confident in their learning, we calculated their mean scores for items P6 and C6 (Table 14). The mean score of the low-scoring group for items P6 and C6 was 4.000 (which is a high value).

**Table 14.** Mean score of the low-scoring students for items P6 and C6.

| NO | Item | Average |
|---|---|---|
| P6 | Our group could complete the assignments on the worksheets within the stipulated time. | 4.000 |
| C6 | I participated in discussions within the group. | 4.000 |

## 5. Discussion and Conclusions

The paired sample *t*-test revealed that significant differences existed in learning outcomes among the experimental subjects in the experiment; thus, H1 was supported. Further analysis of the learning outcomes revealed that after the experimental and control groups were subjected to the teaching strategy experiment, the experimental group exhibited significantly higher improvements in learning outcomes than did the control group; thus, H2 was supported. The statistical analysis of the learner categories of the experimental group indicated that no significant differences existed between the learning outcomes of the high-scoring students before and after the implementation of the proposed teaching strategy. This phenomenon was possibly observed because these members had high learning ability and could still perform well when subjected to heterogeneous grouping. The medium-scoring students exhibited improvements in grades but not in learning outcomes, before and after the implementation of the proposed teaching strategy. However, significant differences existed between the learning outcomes of the low-scoring students before and after the implementation of the proposed teaching strategy. This phenomenon was possibly observed because the low-scoring students learned under heterogeneous grouping, where some high- and middle-scoring students guided the learning of the low-scoring students and engaged in discussions with them. The aforementioned results indicate that H3 and H4 were not supported but H5 was supported.

The mean values of 3.815, 3.948, and 4.053 obtained for the dimensions of flipped classroom, PBL, and collaborative learning, respectively, indicate that H6 was supported. Among the six items of the flipped classroom dimension, the highest mean was obtained for F2 (i.e., "The flipped classroom model enhanced my active learning ability.") The mean score for this item was 4.000, which indicates the high acceptance of the flipped classroom model by the participants. The lowest mean among the aforementioned six items was obtained for F3 (i.e., "I previewed the class content before class."). The mean score for this item was 3.355, which indicated that the learners were not highly willing to engage in pre-class study. This phenomenon was observed presumably because the aforementioned activity was not a part of the learners' previous learning habits and required additional time spent studying outside of class hours. Among the eight items of the PBL dimension, the highest mean was obtained for P6 (i.e., "Our group could complete the assignments on the worksheets within the stipulated time."). The mean score obtained for this item was 4.113, which indicates that the participants could complete the tasks on the worksheets in the stipulated time through group work. The lowest mean among the aforementioned eight items was obtained for P2 (i.e., "The worksheets helped me understand what I was learning."). The mean score obtained for this item was 3.871, which indicates that the content of the problem-oriented worksheets had to be strengthened in terms of problem descriptions and progression steps. Among the seven items of the collaborative learning dimension, the highest mean was obtained for C6 (i.e., "I participated in discussions within the group."). The mean of 4.274 obtained for this item indicated that the participants were willing to engage in discussions in the group activities. Among the aforementioned eight items, the lowest mean was obtained for C3 (i.e., "Through group work, I could understand the learning content faster."). The mean score of this item was 3.984, which indicates that



some participants could not learn quickly when participating in group learning possibly because of the differential abilities of group members.

Table 12 indicates that the mean scores of the control group were higher than those of the experimental group for items P6, C5, and C6. This phenomenon was observed because the learning abilities of the low- and medium-scoring students of the experimental group were lower than that of the high-scoring students of this group. Thus, the high- and medium-scoring students may have had to help the low-scoring students, or the high-scoring students may have had to help the middle- and low-scoring students in understanding the learning content. Thus, the experimental group spent more time assisting each other in completing worksheet assignments together than did the control group. In addition, the low-scoring students exhibited poor programming skills, which made it difficult for them to understand and keep up with discussions. This difficulty affected their willingness to participate in discussions. The mean scores of the experimental group for items Q7, C4, and C7 were higher than those of the control group for these items. These results were probably obtained because the experimental group comprised members with different abilities who engaged in group work. Therefore, compared with the control group, the experimental group had more ideas for obtaining solutions to the problems during the joint discussion; thus, the experimental group could complete the relevant work correctly. Moreover, because the experimental group had high programming skills, they preferred to work together to solve problems. We analyzed the learning motivation of the low-scoring students in the experimental group and found that the proposed teaching strategy helped them trigger their learning motivation and accomplish learning goals together, through collaborative group learning and participation in group discussions (Table 14).

In conclusion, the combination of PBL and collaborative learning strategies with flipped classrooms enhances the learning outcomes of learners, which is consistent with the results of previous studies. The experimental group, which comprised of heterogeneous subgroups, exhibited significantly higher learning outcomes than did the control group. In addition, the learning outcomes of the low-scoring students of the experimental group improved significantly after the implementation of the proposed model. Thus, heterogeneous grouping achieves a superior learning outcome to homogenous grouping in collaborative learning. The results of questionnaire analysis indicated that the learners were highly satisfied with all aspects of the three teaching strategies: flipped classroom, PBL, and collaborative learning. Thus, the learners were willing to accept nontraditional teaching methods. They completed knowledge transfer before the lesson and internalized their knowledge through cooperative group discussions during the lesson to solve the problems in the PBL worksheets. The adopted teaching method created a pleasant learning environment, where learners no longer had to tolerate the boring learning environment of traditional face-to-face classrooms, in which instructors deliver the learning content, and students listen as an audience. In the teaching experiment, the learners learned individually and through group discussions to solve the problems in the PBL worksheets. They worked hard to accomplish personal and group learning goals, which triggered their learning motivation.

Because of constraints related to time, personnel, and teaching context, this study has certain limitations. First, the participants of this study are students from our university's Department of Information Management. Thus, the results of this study may not be generalizable to students from other departments or universities. Follow-up research can be extended to regional universities or even universities across all of Taiwan to validate the findings of the present study and obtain detailed quantitative data to expand the depth of research. Second, the experimental teaching content was designed with reference to the modules in the "New Concept Visual C# Programming Sample Textbook;"; thus, the inferential nature of the teaching content is limited. Third, in the teaching experiment, heterogenous subgroups comprising students with different programming abilities were created for collaborative learning. The interpersonal and psychological characteristics of the students were not considered. If the factors that negatively affect students' learning can be eliminated, students would be able to cooperate and learn with each other during

group activities without being forced to form certain groups. Fourth, the collaboration subgroups were guided to work together on an outcome through a worksheet; however, the differences in the functioning of the subgroups were not examined. Future research can examine the differences in functioning between discussion groups when implementing the teaching strategy proposed in this study. In addition, qualitative interviews can be added to the research methodology. The interview results obtained from an experimental group can be cross-referenced with those obtained from a control group to gain a deeper understanding of the problems faced by students in the collaborative learning process so that the factors that negatively affect student learning in this process can be eliminated.

**Author Contributions:** Conceptualization, Y.-H.C.; methodology, Y.-H.C., Y.-T.L.; software, Y.-C.Y.; validation, Y.-H.C., Y.-C.Y. and Y.-T.L.; writing—original draft preparation, Y.-H.C. and Y.-C.Y.; writing—review and editing, Y.-H.C. All authors have read and agreed to the published version of the manuscript.

**Funding:** This research was partly funded by Ministry of Education, Taiwan, grant number 34001100218-EDU.

**Institutional Review Board Statement:** The study was conducted in accordance with the Declaration of Helsinki.

**Informed Consent Statement:** Informed consent was obtained from all subjects involved in the study.

**Data Availability Statement:** Not applicable.

**Conflicts of Interest:** The authors declare no conflict of interest.

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
