# Peer review of "Effects of Combining Different Collaborative Learning Strategies with Problem-Based Learning in a Flipped Classroom on Program Language Learning"

_sustainability, doi:10.3390/su14095282_

Round 1
Reviewer 1 Report
- This manuscript proposed combining problem-based learning with different collaboration learning strategies in flipped classrooms to improve learners’ learning motivation and learning outcomes. Some positive results are obtained from the experiment.
- The different collaboration learning strategies are applied in class to improve the learning motivation and learning outcome of then learners.
- As presented in Table 8, the learning outcomes of the low-scoring students improved significantly after implementing the proposed teaching strategy. Therefore, the authors showed that the learning outcome of the heterogenous subgroups is prior to the randomly subgroups.
- The results of this manuscript are of reference value.
- The authors could check the manuscript format again to meet the requirement.
Reviewer 2 Report
The article is well organized. The topic is interesting for anyone interested in the problems of teaching and learning in general and in the initial teaching and learning of programming.
The article makes a review of works with good results in the area, but also on the disadvantages or problems of its use.
The article presents and analyzes the results obtained from a practical study.
The article is interesting and suitable for publication.
to review:
- In the title: review the use of the word Program by Programming.
- Line 63: consider reviewing the way in which authors are indicated. Although correct and consistent with the one used, this one in particular, special attention should be given. Suggestion: “Fetaji et al [7]…”
- Line 118: Consider including a new paragraph for the beginning of the new Problem-based learning (PBL) topic.
- Figure 1: improve figure. The study deserves a better and representative picture of what is described.
- Line 180 and 183: sections too short to be used as a section. It is suggested to combine the two.
- Figure 2: place all the boxes the same size. Uniform words with capital letters.
- Figure 3: according to the flow represented, review the use of the symbol decision flowchart in programming implementation.
- Lines 258, 260, 283: review usage and formatting.
- Figure 4: “Lotto Genertor” ??? or Generator
- Line 373: the value 67.0 does not agree with table 6. It should be 70.0
- Line 385: 57.54 does not agree with table 57.53
- Uniform all table values ​​with the same number of decimal places.
- Line 412: 4.035 does not agree with table 4.053
Author Response
Please see the attachment.

This manuscript is a resubmission of an earlier submission. The following is a list of the peer review reports and author responses from that submission.